# Assessing Related Factors of Intention to Perpetrate Dating Violence among University Students Using the Theory of Planned Behavior

**DOI:** 10.3390/ijerph17030923

**Published:** 2020-02-02

**Authors:** Wen-Li Hou, Chung-Ying Lin, Yu-Ming Wang, Ying-Hua Tseng, Bih-Ching Shu

**Affiliations:** 1College of Nursing, Kaohsiung Medical University, Kaohsiung 80708, Taiwan; wlhou422@gmail.com (W.-L.H.); yhtseng@kmu.edu.tw (Y.-H.T.); 2Department of Medical Research, Kaohsiung Medical University Hospital, Kaohsiung 80756, Taiwan; 3Department of Rehabilitation Sciences, The Hong Kong Polytechnic University, Hung Hom, Hong Kong; cylin36933@gmail.com; 4Department of Psychology, Chung Shan Medical University Hospital, Chung Shan Medical University, Taichung 40201, Taiwan; wym@csmu.edu.tw; 5Department of Nursing and Institute of Allied Health Sciences, College of Medicine, National Cheng Kung University, Tainan 70101, Taiwan

**Keywords:** university student, dating violence, structural equation modelling (SEM), intention, theory of planned behavior (TPB).

## Abstract

Dating violence (DV) is a major public health problem among youth. The majority of DV studies in Taiwan involve the assessment of DV without the use of a robust psychological framework to guide research accuracy. Therefore, the purpose of this study was to utilize the Theory of Planned Behavior (TPB) to assess intention and other salient factors related to DV among university students. A two-phase, mixed-method design study was conducted among university students from seven universities in Taiwan, aged 18 to 24 years. In Phase I, questionnaires used were specifically developed based on the TPB, consisting of both direct and indirect measures related to DV. In Phase Ⅱ, questionnaire evaluation and related factors were examined through a two-step process of structural equation modelling (SEM) to test the TPB model. The results of this study found that perceived behavioral control, subjective norm, and attitude toward DV on behavioral intention accounting for 37.5% of the total variance. Among the related factors, attitude toward the behavior was the strongest (β = 0.48, *p* < 0.001), followed by perceived behavioral control (β = 0.19, *p* < 0.05). Findings from this study could expand the knowledge base in this important area and might help prevent DV.

## 1. Introduction

Dating violence (DV) is a global phenomenon and public health problem affecting approximately 29% to 59% of young males and females [1,2,3]. DV includes four types of behavior, namely, physical violence, sexual violence, psychological abuse, or stalking, and occurs between two people in a close relationship [4]. Males and females may be different in the DV status (victim, perpetrator, or both) within traditional in-person and cyber abuse [5]. Though females disclosed higher levels of battering than males [6], suicide attempts among male victims of sexual DV were greater than those of female victims [7]. Most victims undergo multiple forms of violence [8,9,10], but they seldom seek help [3]. Moreover, DV has been proven to cause serious short- and long-term negative consequences to individuals, such as physical injury [11], suicide idea or behavior [12], and homicide [13]. 

Early adulthood (aged 19–30 years) is a critical stage in which humans develop close, committed, and intimate relationships with other individuals, according to Erik Erikson’s theory of psychosocial development [14]. Women aged 16 to 24 who were the most likely to be in an active dating relationship were at the highest risk of nonfatal intimate partner violence [15]. Moreover, early DV was highly correlated to adult marital violence [16], which is more serious because it disrupts family harmony and is a risk factor for children who witness violence to become perpetrators or victims of intimate partner violence as adults [2,17,18]. The high incidence rate and devastating consequences of DV for individuals and social development are reflected in the increased demand for preventive services among youth. Efforts to develop specific preventive interventions against DV must be increased [19].

In Taiwan, few studies have examined how perpetrators use violence against their dating partners, although there are some available questionnaires on the type and risk factors of DV [1,3] and attitudes toward DV [20,21]. According to social cognitive theory [22], the behavior of an individual is determined by the interaction with his/her cognition, other personal factors, and environmental events. Previous research on the related factors and predictors of DV perpetration among young men and women reported that misconception of DV [23,24] and belief in gender stereotype [21,25,26] may influence the attitude of accepting DV. Moreover, a meta-analysis conducted by Park and Kim (2018) concluded 131 related factors of DV perpetrators among 27 studies. Their results further showed that “family relationship problems” and “having deviant peers” were the strongest risk factors of DV perpetration in family-related factors and community-related factors, respectively [27]. However, DV perpetration is not well understood/investigated among Taiwanese university students because limited research has examined the socio-cognitive determinants of intention to perpetrate DV. Hence, the application of a robust psychological framework in DV needs further studies to support. To address these gaps, the current study attempted to identify comprehensive socio-cognitive determinants (e.g., family and peers, gender stereotypes, misconception of DV) of gender-based violence that underlie DV behaviors of perpetrators with related behavioral theories. 

A key point in developing effective interventions preventing DV behaviors among youth is to gain a thorough understanding of DV intention processes and theoretical underpinnings. The Theory of Planned Behavior (TPB), a model rooted in social psychology [28], might help in the analysis of DV, because it considers the interaction between personal and social factors to explain behavior [29]. TPB has been used in the field of prevention [30] and can successfully predict and explain various behaviors and intentions [31,32]. From the perspective of primary prevention, TPB has been applied to numerous areas of research on violence, including the prediction of cyberbullying [33], reporting forced sex [34], and fighting [35]. 

TPB is an extension of the Theory of Reasoned Action [36,37], according to which specific behavioral intentions are the determinants of behaviors. Intention is defined as an individual’s motivation towards a specific behavior and is determined by both attitude toward the specific behavior and subjective norm regarding the behavior. The Theory of Reasoned Action assumes that human behaviors are fully under the volitional control of individuals. However, sometimes human behaviors which are beyond the control of individuals’ volition are also influenced by strong emotions or others’ behavior. Ajzen (1985, 1991) proposed the TPB, which added the predictive construct of perceived behavioral control to the Theory of Reasoned Action. Perceived behavioral control is defined as the perceived ease or difficulty of performing a specific behavior [29,38]. Hence, explanation of behavioral intention might be enhanced by considering not only attitude and subjective norm, but also perceived behavioral control. These three determinants are indirectly influenced by the individual’s salient beliefs, including behavioral beliefs, normative beliefs, and control beliefs [39]. In general, the more the positive attitudes, subjective norms, and perceived behavioral control towards the particular behavior, the stronger the individual’s intention towards the behavior [29,38]. The associations among the TPB factors (i.e., attitude, subjective norm, perceived behavioral control, and behavioral intention) indicate that changing attitudes, subjective norms, or perceived behavioral control may be associated with changes in behavioral intentions and the final behavior [39,40,41,42,43,44,45,46].

Empirical validations of the TPB have shown that the model explains 16.4–44.8% of the variance in intention which is related to violence [33,34,47]. Amar (2009) supported the utility of TPB in exploring the association of intention to report forced sex with attitudes and beliefs by African American women in a university and proved it to be a useful framework [34]. Heirman and Walrave (2012) demonstrated support for the relationships among the theoretical constructs of TPB based on findings from the study of adolescents’ self-reported perpetration in cyberbullying and reported that the predictor variables (attitude, subjective norm, and perceived behavioral control) accounted for 44.8% of the variance in intention, and attitude was the strongest predictor, followed by perceived behavioral control and subjective norm [33]. However, a study by Ma et al. (2017) showed that only attitude was the significant predictor of nurses’ negative perceptions to being bullied in the workplace [47]. Although there were different research results, the strength of the TPB provides a useful theoretical framework to decode individuals’ actions and allows the investigator to capture the motivational factors/intentions that influence behavior [38]. However, there is a lack of validated TPB measures tailored specifically to DV behavioral intention.

The above review suggests initial support for the TPB-based screener for university students, and indicates a need for additional research in particular. Hence, the purposes of this study were as follows: (1) to develop a valid and reliable questionnaire, the Dating Violence Behavioral Intention Questionnaire (DVBIQ), to assess the intention to perpetrate DV within the TPB framework; and (2) to explore the related factors of the intention to perpetrate DV among university students in Taiwan. Based on theoretical [39] and empirical evidence [33,34,47], we hypothesized the following: (1) positive attitudes towards DV perpetration (i.e., considering that DV perpetration is acceptable) increase the intention towards DV perpetration (i.e., willing to perform DV perpetration). (2) Positive subjective norms towards DV perpetration (i.e., the important others of an individual accept and encourage DV perpetration) increase the intention towards the DV perpetration. (3) Greater perceived behavioral control towards DV perpetration (i.e., feeling that it is easy to perform DV perpetration) increases the intention towards DV perpetration.

## 2. Materials and Methods

### 2.1. Study Design and Participants

A two-phase, mixed-method design was adopted. Phase I involved developing a questionnaire based on the TPB. Phase II conducted psychometric validation and associations of the TPB-based model. The participants were eligible if they were Taiwanese university students between 18 and 24 years, could communicate in either Mandarin or Taiwanese, and agreed to participate in this study. University students who were married or unable to complete the questionnaires were excluded.

Because the most complicated analyses used in this study were confirmatory factor analysis (CFA) and path analysis via structural equation modelling (SEM), we used the two analyses to estimate required sample size. In the CFA, the degrees of freedom are 98; with the setting of type 1 error = 0.05, power = 0.8, null root mean square error of approximation (RMSEA) = 0.08, and alternative RMSEA = 0, the estimated size is near 67. In the path analysis, the degrees of freedom are 22; with the same setting of CFA, the estimated size is near 163 [48]. Thus, the sample sizes in each phase of the present study were sufficient. All participants (n = 465) in the two phases were independent, and finally, the number of participants who were involved in the study were 150 in Phase I and 269 in Phase Ⅱ. For detailed information please see Table 1.

### 2.2. Data Collection

#### 2.2.1. Phase Ⅰ: Questionnaire Development

The questionnaire was developed as proposed by Ajzen and Fishbein (1980) and Ajzen (1988) [36,49] and divided into direct and indirect (belief-based) measures [38]. Direct measures were used to obtain a general assessment or opinion of the participants. Indirect measures served to examine the participant’s underlying specific beliefs and outcome evaluations (Figure 1).


***Step 1: Item development.***


A semi-structured qualitative interview was used to obtain data from a purposive sampling of students in a university. All participants saw the flyer and actively contacted the first author. The first author assessed participants if they fitted the included criteria and informed the purpose and process of this study, the rights and interests of the participants, as well as the resources if they were needed. After obtaining their written informed consent, the first author conducted a face-to-face interview. Participants who self-reported that they had ever perpetrated violence or mutual violence against a dating partner or ever had serious conflicts with their dating partner completed open-ended questions seeking their opinions and feelings on DV. The questions were framed according to the definitions of the TPB constructs of attitude, subjective norm, perceived behavioral control, and intention to perpetrate DV. Sample questions were “*what do you think are the advantages/disadvantages of dating violence*,” “*what is the perception of significant groups or individuals who approve/disapprove of your committing dating violence against your partner*,” and “*what factors, circumstances, or opportunities would facilitate or be a barrier to your committing dating violence towards your partner.*” Content analysis was performed by two researchers. The interview responses from two males and eight females (n = 10) aged 18 to 22 years were labeled and listed in order of frequency into three categories: attitude, subjective norm, and perceived behavioral control toward DV. Instrument indicators were developed based on the results of the content analysis. Forty items were then combined to produce the first draft of the instrument.


***Step 2: Content validity and face validity.***


A panel of four experts in intimate partner violence confirmed content validity and readability of the preliminary instrument over a two-round process. The content validity of the preliminary instrument used the following four responses: 1 = not relevant; 2 = slightly relevant, with major revision needed; 3 = relevant, with minor revision needed; and 4 = very relevant and concise. The content validity index (CVI) of the questionnaire was calculated by dividing the total number of items with an average expert rating of 3 or 4. Based on the suggestions of the panel, one item was deleted from the attitude subscale, two from the intention subscale, and 20 items were reworded due to irrelevance or redundancy. This questionnaire has a CVI of 0.93, which exceeds the minimum threshold of 0.80 required to indicate its acceptability for further testing [50]. Next, five first-year university students (2 males and 3 females, aged 18–20 years) were surveyed to assess the face validity of these items to assess clarity, readability, wording or formatting, and time to complete.


***Step 3: Item analysis and internal consistency.***


The quantitative pilot study was conducted to test the initial item statistical analysis. Data were collected from a university located in southern Taiwan with a purposive sampling of 150 university students. Finally, 135 (90%) questionnaires were completed by 58 males (43%) and 77 females (57%), with a mean age 21.15 ± 1.44 years, and aged 19–24 years. A minimum of 80 participants in a pilot study to test the psychometric properties of direct and indirect measures was recommended by Ajzen (2006) [39]. Six items were deleted from the perceived behavioral control subscale due to the corrected item-total correlation being less than 0.2. The Cronbach’s alpha for the six latent constructs in the direct and indirect measures was from 0.62 to 0.92 and all were considered reliable (Cronbach’s alpha >0.60) when the questionnaire included less than 10 items [51]. The pilot study resulted in a 31-item first-version DVBIQ (9-item direct measures and 22-item indirect measures) divided into four subscales: attitude (11), subjective norm (8), perceived behavioral control (11), and intention (1).

#### 2.2.2. Phase Ⅱ: Psychometric Validation and Predictability of the TPB-Based Model

A stratified random sampling was adopted to recruit the participants. From Kaohsiung and Pingtung in southern Taiwan, the universities were randomly selected according to the ratio of university type (national university: private university = 2:4) and then the departments of the university. Subsequently, selected departments including four classes (first, second, third, and fourth grade) were invited to participate in the study. Eight universities were contacted, of which two universities refused to take part in this study because the fourth-grade university students conducted internships outside the school or the class time was limited and the questionnaires could not be answered. Participants (n = 300) who volunteered to participate were asked to fill out a structured questionnaire. These sample sizes met the minimum requirement of 5–10 samples per item to support the factor analysis [52].


***Step 4: Validity and reliability validation.***


A cross-sectional study was conducted in southern Taiwan from April to June 2017 with a sample of 23 universities: 8 national universities and 15 private universities. The final version of 26-item scale, the Dating Violence Behavioral Intention Questionnaire (DVBIQ), was validated with construct validation, applying CFA, path analysis via SEM, and internal consistency by McDonald’s omega. Concurrent validity cannot be analyzed because there are no same concepts and variables to measure dating violence behavioral intention in Taiwan.


***Step 5: Criterion-related validity.***


Criterion-related validity of the instrument was tested using path analysis to examine the overall potential of the questionnaire to explain intention and to identify the related factors. For all outcomes, a *p*-value <0.05 was considered statistically significant. The explanatory power of the model was obtained from the R^2^ value of the model.

### 2.3. Instrument

Two measurements were used in this study: general information (e.g., gender, age, year of study in university, current dating partner, and love experience) and the 26-item Dating Violence Behavioral Intention Questionnaire (DVBIQ) composed of four subscales: attitude (7), subjective norm (8), perceived behavioral control (10), and intention (1). The questionnaire was constructed as follows:

#### 2.3.1. Attitude Subscale

This subscale comprised seven items and was divided into two components: direct attitude and behavioral beliefs. To evaluate direct attitude (three items), the opinion of participants was sought on DV, for example, “*If I have a dating partner, my feeling to physically, mentally, or sexually abuse him/her in the following month is….”* Each item was evaluated using a seven-point scale: 1= extremely bad to 7 = extremely good (range 3–21).

The behavioral beliefs were assessed with two composite items including behavioral belief (behavior leads to certain consequences) and outcome evaluation (of the advantages and disadvantages of these consequences). The two behavioral belief statements were graded on a seven-point Likert scale (1 = strongly disagree; 7 = strongly agree) and the two outcome evaluation items (1 = extremely bad and 7 = extremely good). The score of each item was multiplied by its corresponding outcome evaluation, and these product scores were summed up for a weighted belief score (range 2–98).

For both components, higher values meant that participants had favorable attitudes to perpetrating DV.

#### 2.3.2. Subjective Norm Subscale

This subscale comprised eight items and was divided into two components: direct subjective norm (SN) and normative beliefs. To evaluate direct SN (two items), the participants were asked what their perceptions were of social pressure to perpetrate DV, for example, “*Most of the people who influence me a lot (e.g., parents, friends, and teachers) hold an attitude that physical, mental, or sexual violence against a dating partner is…*.” Each item was evaluated using a seven-point scale: 1= strongly disagree, to 7= strongly agree (range 2–14).

Normative beliefs were assessed with three composite items including normative beliefs (normative expectations of specific referent groups) and motivation to comply (with those groups). The three normative belief statements and three motivation to comply statements were graded using seven-point Likert scales (1 = strongly should not, to 7 = strongly should and 1 = very willing to, to 7 = very unwilling to, respectively). The score of each item was multiplied by its corresponding motivation to comply, and these product scores were summed up for a weighted belief score (range 3–147).

For both components, higher scores reflected a greater perception of others expecting participants to perpetrate DV or that DV was accepted as normal or deserved.

#### 2.3.3. Perceived Behavioral Control Subscale

This subscale comprised 10 items and was divided into two components: direct perceived behavioral control (PBC) and control beliefs. To evaluate direct PBC (three items) the participants were asked about their self-efficacy and perception of control in the perpetration of DV, for example, “*To me, physical, mental, or sexual violence against a dating partner in the following month is…*” Each item was evaluated using a seven-point scale: 1 = strongly not easy to 7 = strongly easy (range 3–21).

Control beliefs were assessed by four composite items including controlling beliefs about what facilitates perpetrating DV and the perceived power of these factors, for example, “*If the dating partner has a bad attitude (e.g., irresponsibility, disobedience, perfunctoriness, provocation, lying) that will provoke my physical, mental or sexual violence against a dating partner…* ” and “*If the dating partner has a bad attitude (e.g., irresponsibility, disobedience, perfunctoriness, provocation, lying)*, it *will provoke my physical, mental or sexual violence against her/him. I feel…*” The four control belief statements and the four power of control statements were graded using seven-point Likert scales (1 = strongly possible to 7 = strongly impossible and 1 = absolutely to 7 = absolutely not, respectively). The score of each item was multiplied by its corresponding power of control, and these product scores were summed up for a weighted belief score (range 4–196).

For both components, higher scores reflected the individual having greater perceived behavioral control to perpetrate DV. Appendix A shows detailed scale content.

#### 2.3.4. Behavioral Intention Subscale

Behavioral intention of DV was measured using one question: “*If I have a dating partner, the possibility of my perpetrating dating violence against him/her in the following month is…*” Responses ranged from 1 (very unlikely) to 7 (very likely) (range 1–7). A higher score reflected that university students had a greater intention to perpetrate DV.

### 2.4. Procedures and Ethical Considerations

After we obtained approval to conduct the study from the Human Research Ethics Committee (NCKU HREC-E-105-093-2), the first researcher contacted the administrators of each selected school to obtain their consent for this study. The purpose and procedure of the study and the potential risks and benefits were explained to the students. The students were further informed that anonymity was assured to protect the privacy of the participants. All information related to the study that could be used to identify the participants would remain confidential. The individual participant’s name would remain confidential and could not be identified in the report, and they were free to withdraw from the study at any time, participation being voluntary and anonymous. The data collection process took 10–15 min overall. With the return of one completed questionnaire, participant consent was deemed as given.

### 2.5. Data Analysis

Participants’ responses were entered into SPSS version 22.0 (IBM Corp, Armonk, NY, USA). Descriptive (means and standard deviations) and reliability statistics were conducted, and correlations between study variables were calculated using Pearson Product Correlations. McDonald’s omega was estimated using the Psych package in the R software [53].

Construct validity was assessed by factor analysis. This study used CFA to evaluate the TPB model, because the same is preferred to exploratory factor analysis (EFA) in validation studies when a prior theoretical model is present [54]. IBM SPSS AMOS 24.0 (IBM Corp, Chicago, IL, USA) was used to test the hypothesized TPB model by using a two-step modeling approach recommended by Anderson and Gerbing (1988) and McDonald and Ho (2002): Step 1 is testing the measurement model and Step 2 is testing the structural model (i.e., path analysis) [55,56]. In the first step, CFA was used to confirm and validate the measurement model to assess convergent and discriminant validity [57]. In the second step, path analysis via SEM was used to examine whether the hypothesized TPB model was an acceptable model to the present data. Specifically, the path analysis tested the correlations between any two components under the theoretical framework of the TPB model and examined the total variance of all variables in explaining intention.

## 3. Results

### 3.1. Demographics

For phase Ⅱ, three hundred questionnaires were distributed, and 31 questionnaires were excluded due to a large amount of incomplete data. Finally, 269 (89.7%) university students finished the questionnaire: 119 males (44.2%) and 150 females (55.8%), mean age 20.9 ± 1.3 years, aged 18–24 years, 64.3% had fallen in love with a dating partner, and 41.3% had a current dating partner (Table 2). The sample size fulfilled the requirement of 250–500 for the structural equation modelling, as recommended by Schumacker and Lomax (2004) [58].

### 3.2. Psychometric Properties of the Scales

#### 3.2.1. Internal Consistency and Pearson Correlations among Studied Constructs

McDonald’s omega was used to determine the internal consistency of the subscales created by each factor. McDonald’s omega for the attitude, subjective norm, and perceived behavioral control subscales was 0.94, 0.92, and 0.94, respectively (Table 3). McDonald’s omega more than 0.7, inter-item correlation more than 0.3, and corrected item-total correlation of more than 0.5 for all the subscales showed the DVBIQ has good internal reliability. The McDonald’s omega for the 26-item DVBIQ was 0.97. Moreover, the Pearson correlations showed the following: the relationship between attitude and subjective norm (*r* = 0.44, *p* < 0.001), attitude and perceived behavioral control (*r* = 0.47, *p* < 0.001), as well as subjective norm and perceived behavioral control (*r* = 0.39, *p* < 0.001) were statistically significant positive correlations. However, these four factors were not correlative with age and gender of participants with any statistical significance.

#### 3.2.2. Evaluation of the Measurement Model via CFA

The default maximum likelihood (ML) estimation with AMOS requires continuous multivariate normality of the observed indicator variables. Because multivariate kurtosis represented by Mardia’s coefficient was above the recommended criterion (>3), multivariate normality was violated. To handle non-normality, we used bootstrapping to bias correct estimates and the Bollen–Stine corrected *p*-value instead of ML-based *p*-value to assess model fit [58]. The measurement model was assessed in terms of individual item loadings and reliability of measures, convergent validity, and discriminant validity. Initially, the CFA with seven latent constructs showed factor loadings in direct and indirect measures had nonsignificant loading to their indicator, and factor loadings less than 0.5 were removed from the model. One item was removed from the direct measure (direct PBC), and four from the indirect measure (two items for behavioral beliefs and two for outcome evaluation).

The final set on the DVBIQ comprised 26 items grouped into seven latent constructs. Convergent validity, reflecting the degree of items in each construct, was interconnected by matching their theoretical connection and was based on three criteria: factor loadings > 0.5, composite reliability (CR) > 0.7, and average variance extracted (AVE) for each construct > 0.5 [59]. The results of the re-estimation of individual item loadings and the reliability of measures of the three-factor model are in Table 3. All factor loadings were more than 0.5, and ranged from 0.71 to 0.89, demonstrating good convergent validity at the item level. For CR, all values exceeded 0.7 and ranged from 0.89 to 0.91. Finally, the AVE values were more than 0.5 and ranged from 0.62 to 0.66. The convergent validity for the proposed constructs of the measurement was good because the three criteria (factor loadings >0.5, AVE > 0.5, and CR > 0.6) for each construct were satisfied (Table 3).

Discriminant validity was achieved when AVE values for any two constructs were greater than the squared correlations between the two constructs [59]. In Table 4, the diagonal elements in the matrix are the square roots of the AVE, and because these were higher than the values of their corresponding rows and columns [60] in the measure, discriminant validity appeared satisfactory for all constructs.

Finally, the measurement model of DVBIQ at this stage contained 26 items with eight direct measures and 18 indirect measures and had adequate reliability and construct validity (i.e., convergent validity and discriminant validity) in four domains (attitude, subjective norm, perceived behavioral control, and intention).

#### 3.2.3. Evaluation of the Path Analysis via SEM

SEM was conducted to estimate the fit of the TPB model and the relationships among the latent constructs. A good model fit was indicated by these indices: the ratio of normed chi-square to degree of freedom (χ^2^/df) < 3.0, root mean square error of approximation (RMSEA) ≤ 0.08, Tucker Lewis index (TLI) > 0.9, comparative fit index (CFI) > 0.9, and *p* > 0.05 for the chi-square test [59,61]. Thus, the proposed structural model with the intention to perpetrate DV explained by attitude, subjective norm, and perceived behavioral control was an acceptable model fit to the data, χ^2^/df = 3.358, RMSEA = 0.071, TLI = 0.948, CFI = 0.93, *p* = 0.80. The results of this study support the validity of the TPB as a model for the intention to perpetrate DV among university students.

The path analysis conducted via SEM for direct measures: Figure 2 shows the path coefficients of measures of the TPB structural equation model of the intention to perpetrate DV. Statistically significant positive correlations were observed between behavioral beliefs and direct attitude (β = 0.86, *p* < 0.001), normative beliefs and direct SN (β = 0.78, *p* < 0.001), and control beliefs and direct PBC (β = 1.04, *p* < 0.001) and confirmed that the three indirect beliefs were well constructed. As hypothesized by the TPB, stronger attitudes (β = 0.48, *p* < 0.001) and PBC (β = 0.19, *p* < 0.05) were both linked to stronger intention. However, subjective norm was not statistically significant related to intention (β = 0.08, *p* = 0.197). The final model explained 37.5% of the variance in intention. 

### 3.3. University Students’ Scores for the TPB Constructs

Table 3 shows each item of university students’ mean scores (SD) for the TPB constructs. The total mean scores (SD) of the TPB constructs showed that university students had low favorable attitudes to perpetrate DV (Mean = 8.52, SD = 8.79, Range = 5–61), low perception of general social pressure (Mean = 11.78, SD = 10.82, Range = 5–69), and a slight perceived behavior control (Mean = 29.07, SD = 30.27, Range = 6–157) with a score below the mean scores of the three subscales. Examining behavioral intention, the mean score of intention to perpetrate DV of the sample was 1.29 with a standard deviation of 0.85. The mean score of behavioral intention of the sample fell below the mean score of 3.5 in this study, indicating that a large portion of the samples just have a few intentions to perpetrate DV. Closer examination of the data revealed 40 (14.9%) of the participants responded with uncertainty of intention to be violent toward their dating partner.

## 4. Discussion

### 4.1. Main Findings and Comparison with Literature

The primary aim of this study was to utilize the TPB to understand the DV behavioral intention among university students in Taiwan. A two-phase process led to the development of a reliable and valid instrument for completing this purpose. The current study extends previous application of the TPB by advancing a theory-based model for explaining intention to perpetrate DV. To the best of the authors’ knowledge, this is the first attempt in this realm to assess university students’ intention to perpetrate DV on the basis of the TPB.

The DVBIQ was developed based on the TPB framework and showed acceptable psychometric properties: good internal reliability (three subscales of McDonald’s omega value range between 0.92 and 0.94), good construct validity, and achieved adequate goodness-of-fit indexes evaluated by the CFA and SEM. Consequently, the correlations between the constructs that were hypothesized to be theoretically related were satisfactory, and the scale should be suitable to measure intention to perpetrate DV among university students.

Most of the hypotheses were supported. Specifically, more positive attitudes, higher level of subjective norm, and greater PBC toward DV perpetration were associated with the higher level of university students’ intention to perform DV perpetration. This study affirmed that two independent variables were significant in the association with intention. Among the related factors, attitude towards the behavior was the strongest (β = 0.48, *p* < 0.001), followed by perceived behavioral control (β = 0.19, *p* < 0.05); however, subjective norm (β = 0.08, *p* = 0.197) was not related with behavioral intentions with statistical significance. Collectively, the related factors accounted for 37.5% of the variance in the intention to perpetrate DV in the sample of participants, which was similar to the finding of previous studies that the TPB model explains 16.4% to 44.8% of the variance in intention which related violence [33,34,47]. The findings are consistent with previous studies [28,31], that is, the strongest related factor was attitude; however, the difference between the two earlier studies and this study is that we found that subjective norm was not significantly correlated with students’ intention of perpetrating violence against their dating partner. According to the social cognitive theory [22], behavior, cognitive, and other personal factors and environmental events all operate as interacting determinates that influence each other bi-directionally. Applying the social cognitive theory to TPB, subjective norms can be shaped by environmental/social factors, including family and peers, which had a significant correlation to DV perpetration in the previous studies [62,63]. The inconsistent result may be due to the misconception of DV (e.g., acts of controlling a partner due to jealousy), gender stereotypes [24,26], and lack of education of DV among youth. It is necessary to verify this hypothesis and add other variables (e.g., personality traits, social support) [10,64] to this model to increase explained variance in future research. The results were a reminder that attitudes toward DV and perceived behavioral control should be considered the critical factors in the design of prevention programs targeting DV among university students.

The present study showed that the low score of behavioral intention of the samples indicated a large portion of the samples having a few intentions to perpetrate DV. Only 40 (14.9%) of the participants responded with uncertainty of intention to be violent towards their dating partners; 10.4% were perpetrators of violence among youths during the previous 12 months in a population-based study in Serbia. However, 62.4% of participants had perpetrated violence (psychological violence: 60.6%; physical violence: 24.3%; sexual violence: 6.7%) towards their dating partners during the past year among college students nationwide in Taiwan [3,10]. The reason of low reporting may be due to the shorter investigation period of intention to perpetrate DV (one month) compared with other studies, which sought the DV behavior during one year [3,10]. Evidence showed that the validity of self-reported DV ranges from low to high, depending on the time duration, the type of DV, and gender [3,9]. Moreover, the questionnaire of intention to perpetrate DV has been tested; so it is reasonable to assume that it was sufficiently simple and unambiguous to achieve a satisfactory degree of reliability.

This research is an initial effort to improve the understanding of the DV behavioral intention of university students in Taiwan. The sociocognitive components in the DVBIQ—attitude, subjective norm, perceived behavioral control, and intention to perpetrate DV—have not been extensively studied in Taiwan. Therefore, the findings of this study highlight the importance of attitude and perceived behavioral control in the influence of DV perpetration intention among university students.

### 4.2. Strengths and Limitations

The strengths of this study are adopting the basis of its theoretical approach to examine the intentions of university students in terms of DV and recruiting participants in main study (Phase Ⅱ) by stratified random sampling. However, four limitations exist in this study. First, this study was a cross-sectional study; thus, we were unable to examine the causal relationship between intention to perpetrate DV and actual DV behavior. Further research should be conducted with a longitudinal study (one or two years) to incorporate subsequent measures of behavior, to measure the predictive effect of the intention to perpetrate DV on actual DV behavior. Second, cyber dating abuse is an emerging problem [65,66]; however, this study lacked a survey of the relationship between DV in the offline and online context. We suggest that the differences of DV between offline and online context can be explored, as well as the associations between gender norms and cyber dating abuse in youths in future research. Third, some demographic information may be associated with intention to perpetrate DV (i.e., socio-economic status of family); however, this study lacked a survey of the demographic factors. We suggest the survey of association between socio-economic status of the university students’ families and intention of perpetrate DV in the future study. Fourth, 10 students participated in the individual interview in step 1, and only two were men. It is recommended that the number of male participants can be increased to balance the gender viewpoints in the future study.

### 4.3. Implications for Practice

The present study showed that the related factors of intention to perpetrate DV were attitude and perceived behavioral control. Therefore, attitude towards DV and perceived behavioral control should be considered as important factors when health professionals, teachers, and school administrators design prevention programs targeting DV among university students.

## 5. Conclusions

The current study represented the first attempt to survey the intention to perpetrate DV of university students utilizing the TPB. The results of this study found that attitude was a strong related factor of intention to perpetrate DV, followed by perceived behavioral control. Primary prevention interventions targeting DV at university students should focus on reducing the tolerance and acceptance of DV perpetration, promote gender equality and respect for others, as well as emotional regulation abilities and communication skills when the individual’s needs cannot be satisfied.

## Figures and Tables

**Figure 1 ijerph-17-00923-f001:**
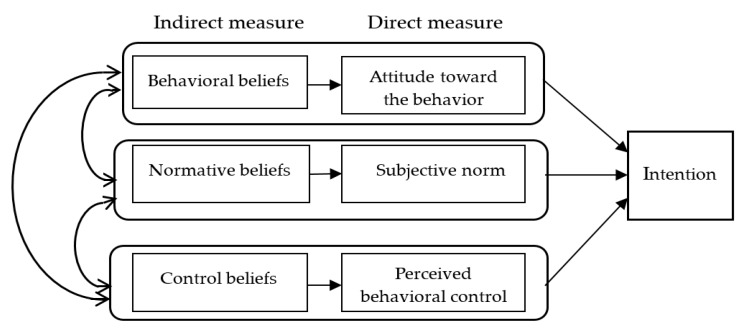
Theoretical framework for intention to perpetrate dating violence based on the Theory of Planed Behavior [39].

**Figure 2 ijerph-17-00923-f002:**
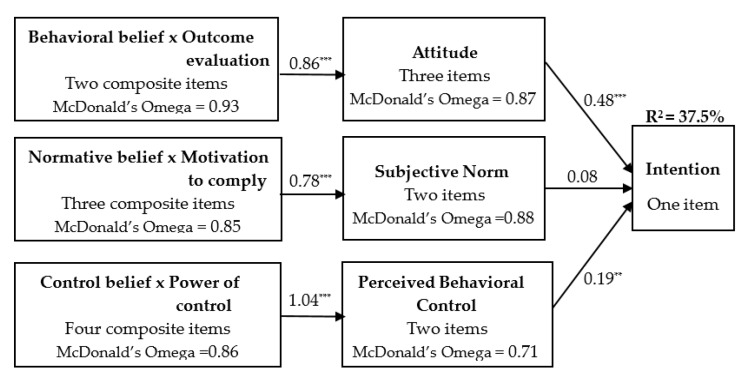
Results for proposed path of the TPB model of the intention to perpetrate DV; *** *p* < 0.001.

**Table 1 ijerph-17-00923-t001:** The processes and samples for developing the Theory of Planned Behavior (TPB)-based questionnaire and main study.

Step	Method (Samples)	Total Items	Description of Samples
Phase Ⅰ: Development Samples came from a university.			
Step 1	Individual interview (n = 10)	40	- A purposive sampling of 10 students. - All participants who saw the flyer and actively contacted the first author: two males and eight females, aged 18–22.
Step 2	Expert panel Face validity (n = 5)	37	- A senior social worker, a consulting psychologist, a criminal justice professor, and a social work professor (n = 4). - Five first-year university students: two males and three females, aged 18–20.
Step 3	Pilot test (n = 150)	31	- A purposive sampling of 150 students. - 135 university students (90%) finished the questionnaire: 58 males and 77 females, aged 19–24.
Phase Ⅱ: Psychometric validation and predictability of the TPB-based model			
Step 4	- Construct validation: confirmatory factor analysis (CFA) and structural equation modelling (SEM) - McDonald’s omega (n = 300)	26 - Attitude (7) - Subjective norm (8) - Perceived behavioral control (10) - Intention(1)	- A stratified random sampling and six universities took part in this study. - 269 (89.7%) university students finished the questionnaire: 119 males and 150 females, aged 18–24.
Step 5	Path analysis	26	

**Table 2 ijerph-17-00923-t002:** Demographics of participants (n = 269).

Variable	Mean (SD)	n (%)
Gender		
Male		119 (44.2)
Female		150 (55.8)
Age (Range: 19–24 years old)	20.9 (1.3)	
19–20 years		119 (44.2)
21–24 years		150 (55.8)
Year in university		
First year		83 (30.9)
Second year		54 (20.1)
Third year		55 (20.4)
Fourth year		77 (28.6)
Current dating partner		
No		158 (58.7)
Yes		111 (41.3)
Had fallen in love with a dating partner		
No		96 (35.7)
1		63 (23.4)
2–4		98 (36.4)
More than 5 times		12 (4.5)

**Table 3 ijerph-17-00923-t003:** Mean, Standard deviation (SD), internal reliability, and convergent validity of the Dating Violence Behavioral Intention Questionnaire (DVBIQ).

Construct	Item	Mean (SD)	McDonald’s Omega	Corrected Item-Total Correlation	Factor Loading (>0.5)	CR (>0.6)	AVE (>0.5)
**Attitude**			0.94			0.91	0.66
Direct attitude			0.87				
	item 1	1.18 (0.60)		0.69	0.80		
	item 2	1.15 (0.47)		0.77	0.85		
	item 3	1.22 (0.59)		0.63	0.78		
Behavioral beliefs			0.93				
	bb1 × oe1	2.51 (3.99)		0.87	0.82		
	bb2 × oe2	2.47 (3.91)		0.88	0.81		
**S** **ubjective norm (SN)**			0.92			0.89	0.62
Direct SN			0.88				
	item 1	1.27 (0.69)		0.61	0.73		
	item 2	1.22 (0.59)		0.69	0.77		
Normative beliefs			0.85				
	nb1 × mc1	2.85 (3.33)		0.73	0.82		
	nb2 × mc2	3.16 (4.07)		0.76	0.83		
	nb3 × mc3	3.29 (3.93)		0.74	0.77		
**Perceived behavior control (PBC)**			0.94			0.91	0.63
Direct PBC			0.71				
	item 1	1.90 (1.46)		0.77	0.75		
	item 2	2.22 (1.60)		0.80	0.88		
Control beliefs			0.86				
	cb1 × pc1	8.08 (10.50)		0.69	0.81		
	cb2 × pc2	7.06 (9.34)		0.74	0.89		
	cb3 × pc3	5.26 (6.82)		0.71	0.72		
	cb4 × pc4	4.55 (6.51)		0.73	0.71		
**Intention**	item 1	1.29 (0.85)			0.96	0.92	0.92

Notes. AVE, average variance extracted; CR, composite reliability. bb—behavioral belief, oe—outcome evaluation, nb—normative belief, mc—motivation to comply, cb—control belief, pc—power of control.

**Table 4 ijerph-17-00923-t004:** Discriminant validity of the DVBIQ constructs.

Construct	AVE	1	2	3
1 Attitude	0.66	**0.81**		
2 Subjective norm	0.62	0.52	**0.79**	
3 Perceived behavior control	0.63	0.43	0.37	**0.80**

Notes. The square root of AVEs are represented in bold, and other values represent correlation.

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
