# Peer review of "Assessing Related Factors of Intention to Perpetrate Dating Violence among University Students Using the Theory of Planned Behavior"

_ijerph, 2020, doi:10.3390/ijerph17030923_

Round 1

Reviewer 1 Report

Introduction:

Literature review is insufficient. I aggree with authors that research considring the TPB elements is limited. However, I cannot agree wit the statment that previous research do not have a robust psychological framework and do not analyze socio-cognitive predictor (page 2, lines 54-61). Indeed, there is recent research analysing socio-cognitive predictors. I suggest to conduct a more complete review of the current literature on dating violence and the broader construct: gender violence.

Hypothesis:

Given the cross-sectional nature of the study is not possible to write about predictors. Authors should write about "associations".

Hypothesis should be more concrete. Authors should specify the direction expected for each element in the TPB theory and explain the grounds of the directions expected.

Method:

In my opinion, the principal problem of the article is the insufficient description of the sample. Author/s should include a discussion of the desired sample based on a power analysis, then the procedure used (i.e., who was contacted about participation), and finally the number of participants who were involved in the study.

Is there any information about the socio-economic status of the adolescent's families? Given the sensitivity of the questions how was anonymity and confidentiality conveyed and ensured? How long did the data collection process take overall?

How many participants were involved in the semi-structured qualitative interview. How were they selected? Did participants in the interview also participated in the validation phase of the scale?

Results:

I suggest to calculate the omega to have a more accurate picture of the validity of the scale.

A rationale should be added to explain why concurrent validity was not analyzed with any other psico-social variable or other measures of dating violence.

Discussion:

Future research should be discussed in more detail.

Implications: authors should discuss how attitudes toward dating violence and behavioral control should be target in preventive programs, not only mention the necessity.

Reviewer 2 Report

This topic's relevant to science and society now. Your work is well structured, and it shows a strong base. However, I will explain below questions that it could improve your article:

Interviews: I find that there are few interviews about all of the men. Is it possible to increase the number of the interview? Number of Universities: I don't have clear the name of the participant's universities. Because inside the abstract, say seven and after it says: study. "Eight universities were contacted, of which 2 universities refused to take part in this study ..." The number of participants: I don't have clear the number of participants, I mean, the number of completed questionnaires. Could you explain it better?  References: You should update the reference for the last two years. Conclusions: Could you explain more?

Author Response

This manuscript is a resubmission of an earlier submission. The following is a list of the peer review reports and author responses from that submission.

Round 1

Reviewer 1 Report

Dear Authors

The manuscript titled "Assessing Predictors of Intention to Perpetrate Dating Violence among College Students Using the Theory of Planned Behaviour" approaches a very important social reality that is necessary to know in order to be able to work of preventive form. The personal, family and social impact of DV is very high for those who suffer from it (and for those who exercise it).

In general, it is a manuscript that presents a very positive idea (linking DV with TPB) but its development can be improved (in my opinion and especially for a high level journal such as IJERPH).
Some important improvement ideas (especially at methodological level):

1. Introduction
The introduction addresses the fundamental questions about DV, although I think it would be interesting to delve further into international review study to give a global idea of the problem from the beginning of the manuscript. Likewise, I think it may be an important limitation not to talk at any time about cyber dating, since communication and information technologies have changed this problem and there is a great relationship between the problems in the offline and on line context.

In addition, they briefly comment on the TPB, but this needs more conceptual development as it is one of the vertebral axes of the manuscript. I don't know if no one had worked DV and TPB, but TPB has been used in other violence problems such as bullying or cyberbullying. Perhaps there are similar experiences with which to relate the TPB to problems of violence in other contexts.

Personally, I think it's a big mistake not to reflect the overall objective of the study (and possible specific objectives). Hypotheses must be based on previous literature (and are not) so perhaps research questions should be asked (if the theoretical framework is not improved in this section).

2. On a methodological level, I consider the work to be very confusing.

a) The type of sampling used is not indicated.
b) I believe that the validation process should be approached more clearly and that special attention should be paid to the validity aspects of the instrument content (test specifications table, conclusions of the panel of experts, piloting, etc.) Nothing relevant on this point is indicated.
In addition, there is no indication of the relevance of making an instrument on it in the introduction (which should have been done to maintain the coherence of the study).
c) I believe that the study sample to be extracted from 7 universities is very low for extrapolation of results (not even exploratory). Why is the sample from the academic year 2016? It has been a long time since the data were collected.
d) I miss a point in the method on data analysis. This is important.
e) The questionnaire is not shown at any time, nor as supplementary material, so replication, usability, etc. are very limited in the study. Without seeing the questionnaire it is very difficult for me to accept a paper (normally this is included translated by indicating for each item its properties of the measure...mean, standard deviation, item-total correlation, alpha if the element is eliminated, etc.) This is not the case.
f) I wonder why the authors have not worked only with participants who have manifested problems (of aggression) in DV to know these issues. It should have been explored among those who have performed the behavior, since if this is the first time they have been interrogated on this issue the TPB is meaningless (in my opinion). Most of the sample is not related to DV behavior (you indicate on line 291-292).
g) Currently, it is very important to perform AFCs (and rely on an AFE beforehand) to generate a solid model. Why haven't you performed an AFC of the model with a larger sample? I think this is an important limitation.
h) There is no analysis based on sociodemographic variables and they can play an important role (I think) and this could influence the regression carried out. In a journal such as IJERPH, it would have been necessary to perform a path analysis to test the model and control statistical problems of the regression.

In general, many of the points mentioned affect the results and the discussion of them.
I believe that it is an interesting exploratory work that requires a deep revision (it is my opinion) and that in its current state it does not seem adequate to me to publish. I am concerned about the possible replicability of a study that does not use questionnaires adapted to its language of other languages and that the sample is very low. Additionally, not knowing what sample has been used, and that most are not related to DV are very important limitations.

I am sorry to comment on this news, but I hope that this will help you to improve this version of the manuscript because the work is important and the subject requires our full attention.

Reviewer 2 Report

This paper investigates an interesting topic as Predictors of Intention to Perpetrate Dating Violence among college students using the guidance of a well-known theory. The study is based on a medium-small sample of young students in Taiwan. There are several strengths of the current study such as the development of a new questionnaire. However, a number of conceptual and methodological problems will be attended in order to make the study understandable and readable.  

INTRODUCTION: 

Authors have provided general epidemiological data of DV, but it is important to provide epidemiological data by gender. The same for suicidal ideation related to DV.  

I can't understand some sentences, a review of English and the wording is neededfrom 43 to 44; from 52 to 57. 

Authors indicate that “Moreover, violence in dating relationships might lead to marital violence, which is more serious [13] because it disrupts family harmony, creates an intergenerational cyclical effect, and leads the children who witness violence to become perpetrators or victims of intimate partner violence as adults [2,14,15]”. Most of references cited are cross-sectional studies. Therefore, authors should soften their language, pointing to the relationships between variables and not causes and consequences. 

Authors base the need for their study on the relative absence of studies that “have considered socio-cognitive determinants in the younger adult populations that may be modifiable through preventive interventions. I don't agree with this statementIt is necessary to indicate what these scarce studies are in order to assess the contribution of their study. For example, I consider that there is a very important line of work in relation to the cognitive components (e.g. gender stereotypes) of gender-based violence that should be incorporated into the literature review (e.g. Daigle, L. E., & Mummert, S. J. (2014). Sex-role identification and violent victimization: gender differences in the role of masculinity. Journal of Interpersonal Violence, 29(2), 255-278). A deeper review of socio-cognitive predictors of DV is needed. Then authors should underline the contribution of their study 

The purpose of the study must be reformulated: the three predictors of the intention to perpetrate DV are predetermined by the authors. 

METHODS 

Information about the recruitment of the participants must be provided at each step of the study

Authors said: “The questionnaire was developed with reference to the instructions” (line 97), what they mean? 

Table 1: What do bold words mean? indicate it in a footnote to the table. 

RESULTS AND DISCUSSION 

Only the 27.4% of the variance of college students intention to perpetrate dating violence was explained by the four examined predictors. Authors should give an explanation to this results and refer to other relevant factors that have not been included in the study. 

“The present study showed that the low score of behavioral intention of the samples indicated a large portion of the samples just have a few intention to perpetrate dating violence”. This result should be compared with national data on DV and gender-based violence in the country. If they are very dissimilar they should provide an explanation and some alternative to the problem of social desirability. If the proposed questionnaire is very sensitive to social desirability this is a very important limitation for design prevention programs based on these results. This limitation must be deeper commented. 

Reviewer 3 Report

The manuscript "Assessing predictors of intention to perpetrate dating violence among college students using the theory of planned behavior" is an interesting piece of research that contributes to the field of dating violence. I have got a few suggestions to make, meant to be constructive, so authors can take them into consideration.

Methods: Study design, participants, and instrument psychometrics.

Authors explain that students were recruited from seven universities in Southern Taiwan. Were these seven universities the only ones contacted to take part in the process? or more universities were contacted but only these seven agreed to take part in the study? If so, how many universities were contacted and why did they refuse to take part in the study? Which are the characteristics of these seven universities? Could authors give a brief explanation of the territory? How many students took part in the study? Could authors picture briefly the students' profile? (what degree are they studying, which year are they studying...)

Although the paper is well structured and well written, authors should correct some english grammatical mistakes.